# Surround suppression in mouse auditory cortex underlies auditory edge detection

**Omri David Gilday**[1], **Benedikt Praegel**[1,2], **Ido Maor**[1,2], **Tav Cohen**[2], **Israel Nelken**[1,2], **Adi Mizrahi**[1,2]*

**1** The Edmond and Lily Safra Center for Brain Sciences, The Hebrew University of Jerusalem, Jerusalem, Israel, **2** Department of Neurobiology, The Hebrew University of Jerusalem, Jerusalem, Israel

* Mizrahi.adi@mail.huji.ac.il

## Abstract

Surround suppression (SS) is a fundamental property of sensory processing throughout the brain. In the auditory system, the early processing stream encodes sounds using a one dimensional physical space—frequency. Previous studies in the auditory system have shown SS to manifest as bandwidth tuning around the preferred frequency. We asked whether bandwidth tuning can be found around frequencies away from the preferred frequency. We exploited the simplicity of spectral representation of sounds to study SS by manipulating both sound frequency and bandwidth. We recorded single unit spiking activity from the auditory cortex (ACx) of awake mice in response to an array of broadband stimuli with varying central frequencies and bandwidths. Our recordings revealed that a significant portion of neuronal response profiles had a preferred bandwidth that varied in a regular way with the sound's central frequency. To gain insight into the possible mechanism underlying these responses, we modelled neuronal activity using a variation of the "Mexican hat" function often used to model SS. The model accounted for response properties of single neurons with high accuracy. Our data and model show that these responses in ACx obey simple rules resulting from the presence of lateral inhibitory sidebands, mostly above the excitatory band of the neuron, that result in sensitivity to the location of top frequency edges, invariant to other spectral attributes. Our work offers a simple explanation for auditory edge detection and possibly other computations of spectral content in sounds.

**Data Availability Statement:** All data and code will be available for direct download through the lab's Github. https://github.com/MizrahiTeam/.

**Funding:** This work was supported by an ERC consolidators grant to A.M. (#616063), Israeli

## Author summary

A central computation performed by the auditory system in the mammalian brain is frequency decomposition. As a result, single neurons throughout the auditory system are often characterized by their preference to basic sound frequencies such as pure tones. Here we tested how single neurons in the mouse auditory cortex respond to sound stimuli with a range of bandwidths around specific central frequencies. We found that single neurons respond strongly to specific bandwidths, and that these responses varied in a regular way based on the sound's central frequency. We could explain the nature of these responses using a simple mathematical model that considers the excitatory central

Science Foundation grant to A.M. (#224/17), and the Gatsby Charitable Foundation. The funders had no role in study design, data collection and analysis, decision to publish, or preparation of the manuscript.

**Competing interests:** The authors have declared that no competing interests exist.

response to frequency and the structure of its inhibitory side bands. We found that some neurons in the auditory cortex are best described by their responses to the location of sharp boundaries, or edges, in the frequency composition of sounds. Taken together, our work highlights aspects of neuronal responses in the mouse auditory cortex that go beyond responses to pure tones.

## Introduction

Surround suppression (SS) is a well-established neural computation in the brain [1]. As its name implies, SS results in the suppression of a neuron's firing rate by stimuli that surround its central area of activation. Classically, neurons in sensory systems are described by a tuning curve, which describes the firing rate of a neuron as a function of attributes of stimulus space. SS arises when responses to stimuli outside the regions that normally drive the cell to fire are suppressed. One paradigmatic version of SS has been described already several decades ago in the visual cortex, as the reduction in firing rate in response to oriented bars of increasing length [2].

The vast majority of research on SS has been performed in the visual system. SS has been found to be a property of visual neurons in many species (e.g. mouse, cat, monkey, and human) and in several processing levels of the visual stream including the retina, superior colliculus, lateral geniculate nucleus, primary visual cortex, and association cortices [2–25]. SS has been proposed to play a role in stimulus saliency [26], edge detection [27], figure-ground separation [28], and redundancy reduction [29–31]. However, SS is not unique to vision. Indeed, lateral inhibitory sidebands of tuning curves have been described in all other senses, from olfaction, through somatosensation, to audition [32, 33]. For example, sideband inhibition has been described in different stations along the hierarchy of the auditory system, from the auditory nerve through the brainstem to cortex [34–37]. Nevertheless, we still have only a rudimentary understanding of the computations SS subserves in sensory systems other than the visual system.

Relatively few studies have explored the sensitivity of neurons to spectral bandwidth in ACx [38–40]. These studies often use a small set of stimuli. For example, in mouse ACx, Li and colleagues sequentially described tuning curves of single neurons to pure tones and then presented broadband stimuli (BBS) with increased bandwidth, but only centered at the preferred frequency of each neuron [40]. Probing this stimulus space, they found a subgroup of neurons in ACx whose responses to stimuli with larger bandwidths (centered on the neuronal preferred frequency) were suppressed. This result is similar to findings in the visual system, in which stimuli of increasing size often evoke a reduced response [3, 4].

Here, we tested exhaustively a 2D landscape of stimulus bandwidth and central frequency. We recorded a large number of neurons in ACx that show SS, supporting previous observations. SS was often expressed as a complex pattern of bandwidth tuning that was asymmetrical in frequency space, changing as a function of the central frequency of the sound. Using a model with a small number of parameters we provide a simple mechanistic explanation for such complex patterns and use our findings to argue that SS is a basic mechanism for generating sensitivity to spectral features such as the top frequency edge of the stimulus.

## Results

### Frequency-bandwidth response areas of auditory cortical neurons

We recorded extracellular spiking activity from head restrained, passively listening awake mice, using the high density Neuropixels probes (see Methods). We targeted our recordings to

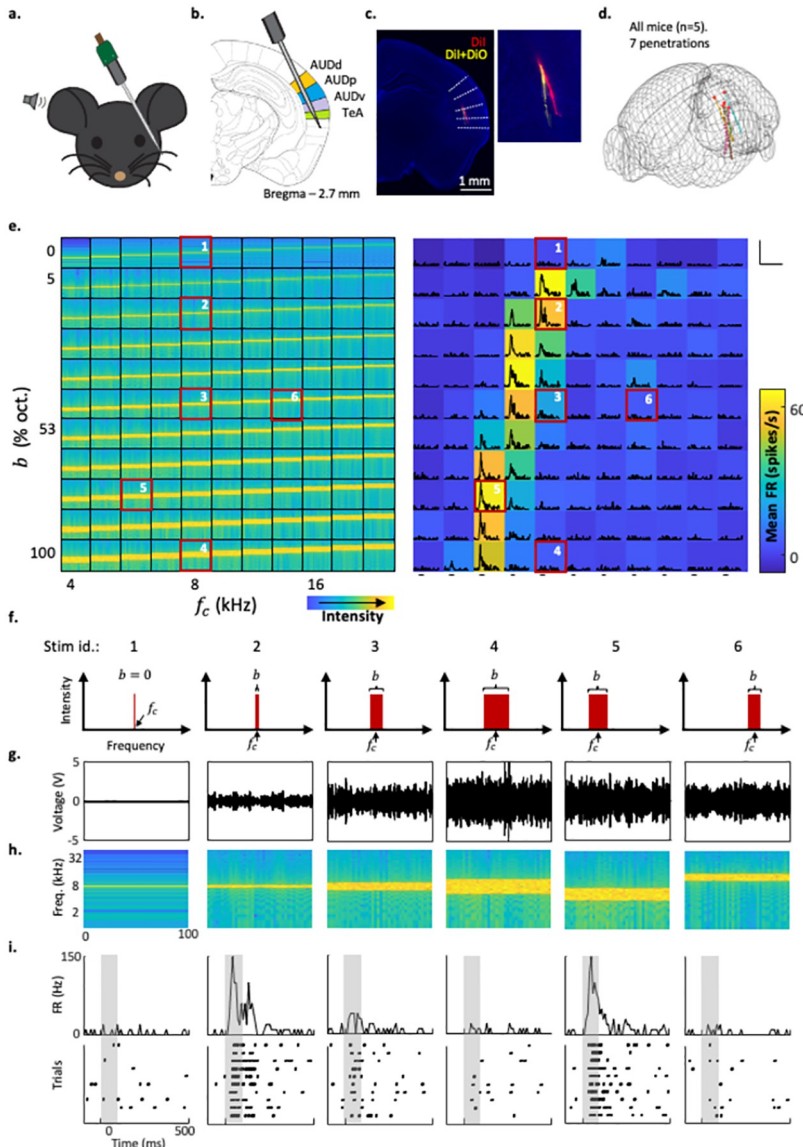

**Fig 1. The experimental design—Recordings and stimuli.** Schematic representation of the experimental setup. a. Probe penetration setting enabling simultaneous recording of activity from several auditory cortices (AUDp, AUDv, AUDd and TeA; highlighted in color). b. Fluorescent image of a coronal brain slice showing two probe tracks in one mouse. c. Reconstructed trajectories of all probe tracks used in this study (n = 7 penetrations; n = 5 mice). d. Left: spectrograms of all presented BBSs sorted based on bandwidth (Y axis, *b*), and central frequency (x axis, $f_c$). Right: A color-coded FBRA of an example neuron in response to all stimuli shown on the left. e. PSTHs are overlayed on top of the mean response at each pixel location. Black bars on the bottom indicate stimulus presentation time. Scale bars— 150 spikes/sec, and 500 ms. f. Schematic of the spectral profiles of the six stimuli marked in e. g. Voltage traces of the six stimuli marked in e. h. High magnification spectrograms of the six stimuli marked in e. i. Top: high magnification PSTHs of the six stimuli marked in e. Bottom: raster plots matching the PSTHs.

the ACx and recorded collectively from 4 auditory regions—primary ACx (AUDp), ventral ACx (AUDv), dorsal ACx (AUDd) and the temporal association area (TeA). To do so, we penetrated the brain with a single probe such that it diagonally traversed all, or a subset of these, cortical regions (Fig 1a–1d). To validate the exact locations of our recordings we reconstructed

**Table 1. Breakdown of cortical areas and layer distribution of the responsive units.**

| Area | layer | Responsive units |
|---|---|---|
| AUDd | 1–4 | 0 |
| | 5 | 1 |
| | 6 | 6 |
| AUDp | 1–4 | 0 |
| | 5 | 22 |
| | 6 | 6 |
| AUDv | 1–4 | 0 |
| | 5 | 32 |
| | 6 | 45 |
| TEa | 1–4 | 0 |
| | 5 | 32 |
| | 6 | 5 |
| Total | | 149 |

(postmortem) the probe tracks, which were coated with a fluorescent lipophilic dye. We used DiI-coated and/or DiO-coated probes for single or multiple sequential penetrations (Fig 1c). All probe trajectories were aligned to the Allen Brain Atlas coordinate framework validating the exact positions of our recordings in all mice (Fig 1d). Using this validation, we found that we collected data almost exclusively from deep layers of ACx (for breakdown of our recording locations see Table 1). Our analysis focused on 149 highly responsive neurons (S1 Fig) from 7 probe penetrations in 5 mice. Spike sorting the data revealed both single units (SUs) and multi-unit data. Here, we present only well isolated SUs strongly responding to the broad band stimuli in the stimulus set (see Methods).

To study the landscape of SS in ACx neurons, we played a two-dimensional array of broadband stimuli (BBS) with varying central frequencies ($f_c$) and bandwidths ($b$) (Fig 1e—left; Fig 1f). We used frequencies between 4-64kHz, 25% of an octave apart with bandwidths of 0–1 octave ($b = 0$ correspond to pure tones; Fig 1e—top row). Sounds (10 trials each) were played in a pseudo-random order of center frequencies and bandwidths. Crucially, we chose to fix the spectrum level of the noise, rather than the total sound intensity of the stimuli. In consequence, sound intensity increased with bandwidth (as measured by the RMS of the signal, Fig 1g). Post-stimulus time histograms (PSTH) were computed for each neuron-stimulus pair (Fig 1e-right; Fig 1i) and used to determine the mean firing rate for each stimulus. Plotting the mean response of all BBS results in a frequency-bandwidth response area (FBRA) for each neuron (Fig 1e-right). Six example stimuli and their responses from the FBRA of the example neuron in Fig 1e (Fig 1e—right; marked in red boxes) are shown in detail in Fig 1i.

FBRAs had heterogeneous shapes. For a given $f_c$, in some cases the responses were a monotonically increasing as a function of bandwidth, while in other cases the responses were non-monotonic, peaking at some bandwidth and decreasing for further increase in bandwidth. Some neurons showed clear asymmetry in their FBRA shapes. Such neurons typically responded only to a small range of bandwidths for any given $f_c$, and the range of effective bandwidths shifted, depending on $f_c$, in a regular manner (see e.g., the six rightmost FBRAs in Fig 2a). Next, we sought to quantitatively explain how such asymmetry could arise. We reasoned that an asymmetric SS would explain the asymmetry detected in the FBRA's.

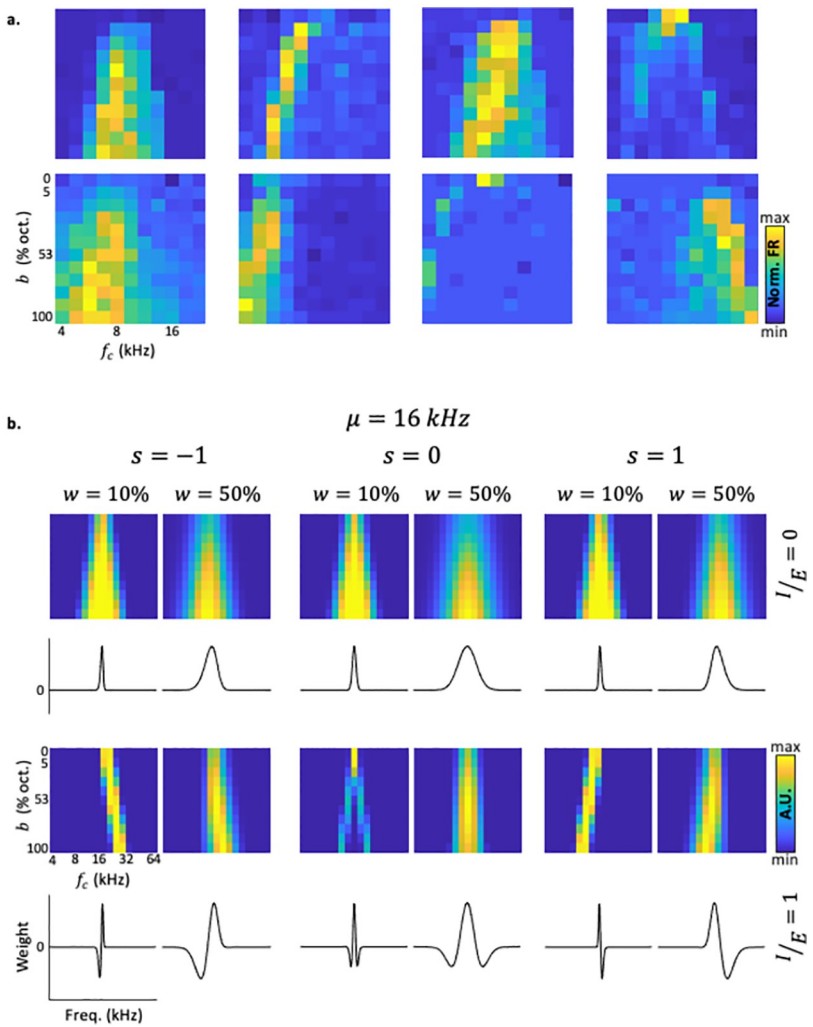

**Fig 2. Frequency Bandwidth Response Areas (FBRAs) of neurons and the model.** a. FBRAs of 8 example neurons from 5 different mice. b. Model-FBRAs (top) and corresponding tuning curves (bottom) of 12 examples as calculated from the model. The examples span varying $s$, $^{I}/_{E}$ and $w$ values.

## Using modulated Ricker wavelets to model asymmetric surround-suppression

In order to explain the FBRAs quantitively, we built a mathematical model, which was inspired by the model that is often used to account for SS in the retina—the Ricker Wavelet (also known as the Mexican Hat) [41]. The Ricker wavelet is the negative normalized second derivative of a Gaussian [42]. The model we used, which we termed the Modulated Ricker Wavelet (MRW) is expressed as:

$$MRW(f) = A \cdot \left(w^2 + {}^{I}/_{E} \cdot (f - \mu)^2\right) \cdot e^{-\frac{(f-\mu)^2}{2w^2}} \cdot \phi^*(f, \mu, w, s)$$

Where $f$ is frequency. The MRW model has four parameters $(\mu, w, {}^{I}/_{E}, s)$. The Ricker wavelet corresponds to $^{I}/_{E} = 1$ and $s = 0$. The mean $\mu$ is a parameter whose only role is to position the wavelet along the frequency axis, and it largely corresponds to the best frequency of the neuron. The width ($w$) determines the overall range of frequencies that affect the output of the

model, through positive or negative weights. The two additional parameters are used to modulate the model. The first parameter is inhibition/excitation ratio ($^I/_E$) which determines the depth of the inhibitory flanks with respect to the excitatory peak. This parameter allows for a smooth interpolation between a gaussian shaped tuning curve that has no inhibitory sidebands ($^I/_E = 0$) to a Ricker wavelet shaped tuning curve ($^I/_E = 1$), which has symmetric inhibitory sidebands (see S1 Video). The second parameter that we added is skewness (*s*). $\phi^*$ is a weighted sum of two sigmoidal functions, adjustable by *s*, that makes it possible to suppress one or the other sidebands, and therefore determines the skewness of the MRW (see Fig 2b and Methods). The parameter *s* has a mild effect when inhibition is weak, and a large effect when inhibition is strong, such that when *s* = ±1, inhibition is present only in one flank of the tuning curve. *A* is a normalization factor that sets the maximal value of the MRW to 1 (Fig 2b; S2 Fig, S1 Video). The MRW is considered as a model tuning curve—a set of frequency-dependent weights that determines how each frequency affects the output of the neuron. The model predicts the responses to a stimulus as a linear sum of the effects of each frequency, with the weights determined by the model tuning curve. By modulating the parameters of the MRW model, we could qualitatively replicate the fine details of many FBRA shapes (compare neuronal FBRAs in Fig 2a to model FBRAs in Fig 2b). For additional details about the model see Methods and S1 Video.

We, therefore, quantitatively fitted neuronal FBRAs to FBRAs computed from the model (i.e. 'model FBRAs'). To do so, we searched for the best possible correlation between neuronal and model FBRAs while adjusting the four model parameters. The result of this search was a single model FBRA for each neuronal FBRA. That is, the model FBRA is the one with the highest correlation between model and neuronal FBRAs. Fig 3a shows four different neurons of well-fitted neuronal FBRAs (Fig 3a—top row), with their best fit model FBRA (Fig 3a—center row), and the resultant model tuning curve (Fig 3a—bottom row). Examples of less successful fits are shown in S3 Fig. At the population level, our model found high correlations between the model and the data, with a mean correlation value of 0.64 ± 0.21 (mean±SD, Fig 3b). These high correlation values dropped to 0.21 ± 0.03 when the neuronal FBRAs were shuffled (Fig 3b; see Methods). These data show that our model has a strong descriptive power for estimating the mean responses of ACx neurons to BBS.

Next, we focused on SUs that had the highest correlations with the model, defined as being above the population median (n = 74 well fitted neurons, S1 Fig; for the distributions of all model parameters that best fit the SU data see Fig 3c). The *μ* values, which are determined by our electrode penetration in relation to the tonotopic axis, were centered at low frequencies (10.5 ± 9.4 kHz; mean±SD), consistent with previous Neuropixels recording experiments from our lab [43]. The mean *w* values were 0.26 ± 0.25 octaves, which are on the order of magnitude of common measures of bandwidth in awake mouse ACx [40] (although *w* here is not precisely bandwidth; see Methods). We then plotted the joint distribution of the parameters *s* and $^I/_E$ for each neuron (Fig 3d). Interestingly, we found that SUs with high correlations to the model belonged mostly to one of two subgroups. One subgroup has typical $^I/_E = 0$ and *s* values close to 0, corresponding to Gaussian tuning curve with no sidebands (Fig 3d—left cluster of points; Fig 3a- left neuron). The other subgroup of neurons tended to have large $^I/_E$ values and a positive *s* (Fig 3d—top right cluster of points). These latter values correspond to a tuning curve with asymmetric sidebands located mostly above the best frequency (FBRAs of such neurons are shown in the two middle examples of Fig 3a). To quantify the distribution of these two subgroups, we divided the graph in Fig 3d to four quadrants, showing that most neurons indeed reside in the top right quadrant (Fig 3d and 3e, 'Q1'). These results suggest that our model explains well the tuning curves of two populations of auditory neurons, one that essentially

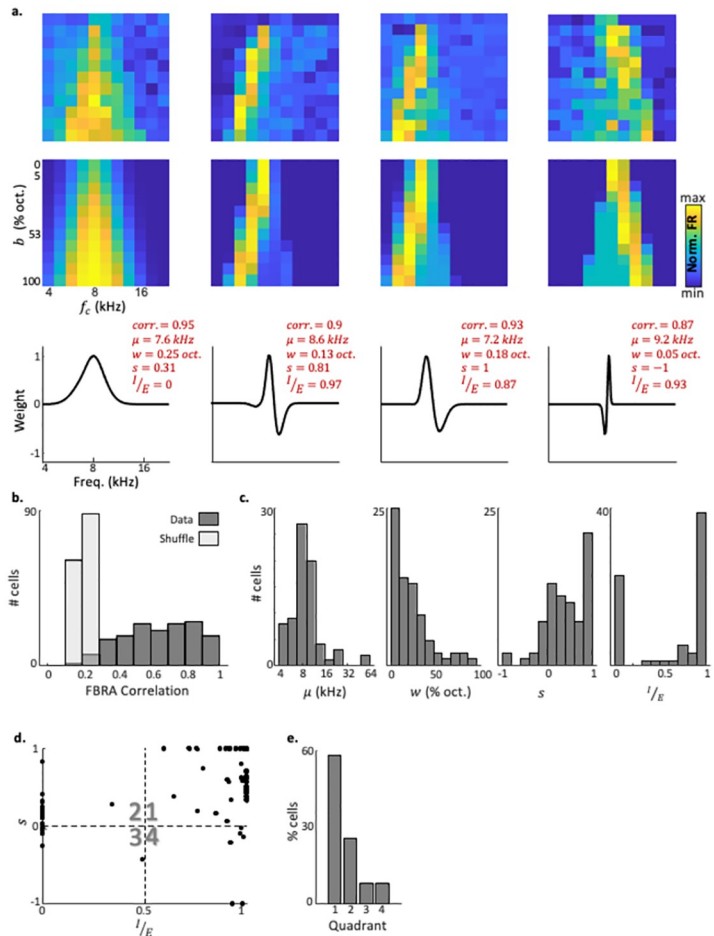

**Fig 3. The MRW model provides a mechanistic explanation to neuronal FBRAs.** a. Top: FBRAs of 4 neurons from 3 mice. Middle: model FBRAs fitted to the same neurons. Bottom: fitted MRWs of the same neurons. b. Distribution of FBRA correlation values (Pearson correlation) for all neurons in our dataset (n = 149 neurons, N = 5 mice) for real and shuffled data. The mean FBRA correlation value was 0.64±0.21 (s.e.m). The mean value of shuffled data is 0.21 ± 0.03. c. Distributions of $\mu, w, s, {}^1/_E$ values of n = 74 neurons. Only neurons with fit correlation higher than the median (which is also 0.64) are shown. d. The combination of skewness and I/E values of n = 74 neurons. We define four quadrants of the graph (Q1-4), such that neurons with high S and I/E are in Q1. The graph shows two main sub-population of neurons in ACx—in Q1 and in Q2. e. Distribution of the number of neurons in each of the quadrant's shown in panel 'd', showing the predominance of neurons with high model fits to Q1.

sums up the energy within its tuning curve (Fig 3d and 3e, 'Q2-3'), and a larger group that has inhibitory sidebands, which are largely above the best frequency (Fig 3d and 3e, 'Q1').

## Model validation using notch stimuli

To test the generalizability of our model, we compared model predictions to a set of stimuli that were not used in fitting the parameters to their actual measured responses. For that purpose, we used notch stimuli, whose frequency content is complementary to that of the BBS. Each BBS (except for pure tones) has a respective notch stimulus (Fig 4a and 4b). Plotting the mean responses to all notch stimuli as a joint function of their center frequency $f_c$ and bandwidth $b$ resulted in a notch FBRA (nFBRA) for each neuron. We used the model with the parameters fitted to the FBRA to calculate a predicted nFBRA for each neuron (Fig 4c, center column). We then compared the measured nFBRA of each neuron (Fig 4c, right column) to its

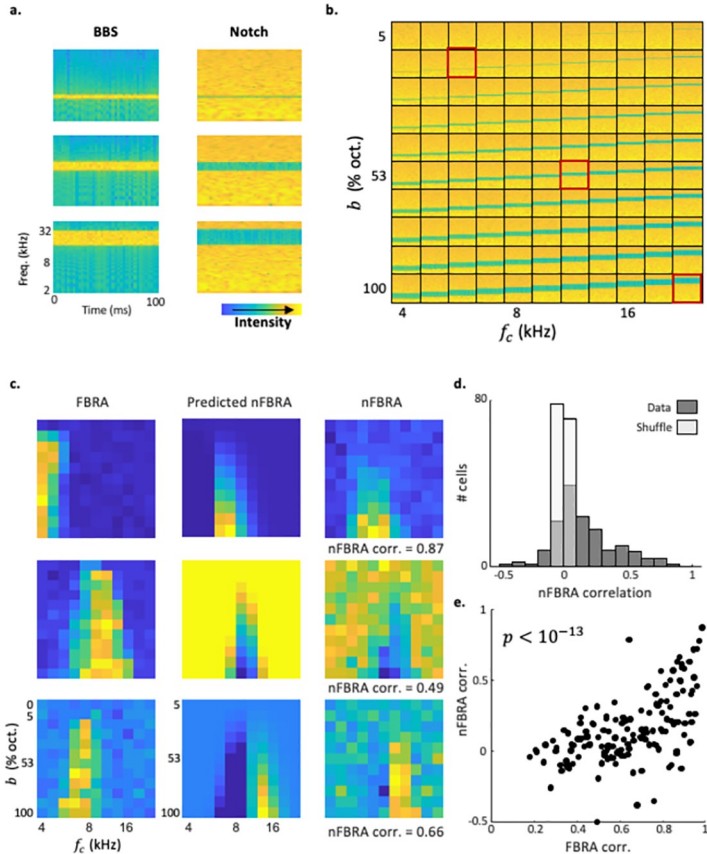

**Fig 4. Validation of the MRW model using notch stimuli.** a. Spectrograms of 3 example BBS (left) and their respective notch stimuli (right). b. Spectrograms of all presented notch stimuli sorted based on inverse bandwidth (Y axis, *b*), and central frequency (x axis, $f_c$). The 3 stimuli from 'a' are marked in red. c. Left: FBRAs of 3 example neurons from 3 different mice. Middle: nFBRAs of the same neurons as predicted by the MRW fit for BBS. Right: nFBRAs of the same neurons. d. Distribution of nFBRA correlation values for real and shuffled data ($p < 10^{-10}$, paired student's t-test). e. nFBRA correlation vs. FBRA correlation for all neurons (n = 149, Pearson correlation, $p < 10^{-6}$).

prediction. The model, which was based on the responses to BBSs, produced predicted nFBRAs that were often highly correlated with neuronal nFBRAs (Fig 4c; Fig 4d compare data to shuffled data). As expected, in neurons for which FBRA correlation values were high, nFBRA correlation values were also high (Fig 4e).

## Neurons in ACx encode the top frequency of broadband stimuli

Finally, we asked what could be encoded by neurons in ACx with tuning curves characterized by high $^I/_E$ and positive **s** values ('Q1' neurons in Fig 3d). Inspired by similar interpretations from the visual system [44], we reasoned that these neurons may respond to edges. Since these neurons are inhibited by higher frequencies rather than lower frequencies, they likely respond only to the top frequency edge, that is, to the top frequency of a broadband stimulus (Fig 5a; marked as $f_t$). Thus, we hypothesized that Q1 neurons will respond largely similarly to broadband stimuli that have the same $f_t$, rather than the same center frequency or the same bottom frequency or the same total energy (Fig 5a and 5b; S2 Video). To test this hypothesis, we replotted the responses to BBSs (Fig 5c, left) as a function of the $f_t$ of the BBSs (Fig 5c, Center). These tuning curves, which were based on top frequency were often unimodal. To quantify this

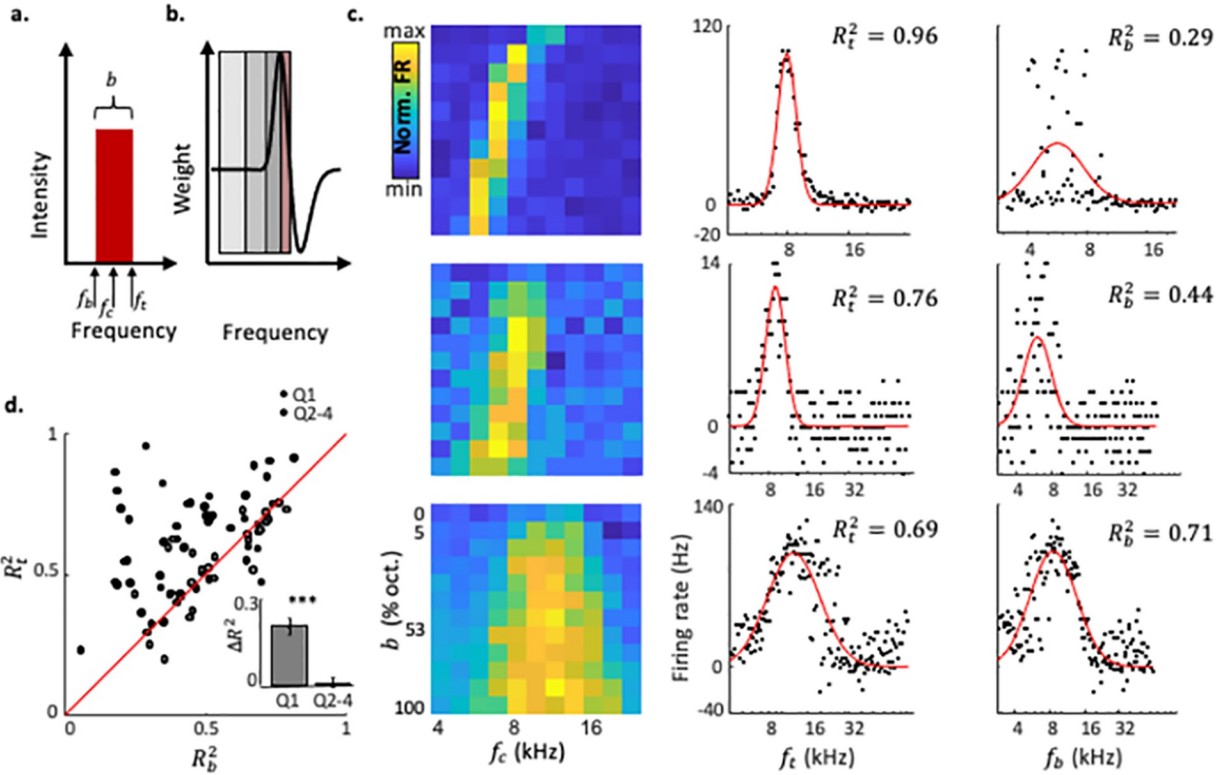

**Fig 5. Cortical neurons selective for top-frequency.** a. Illustration of a BBS. The bottom, center, and top frequencies are marked by arrows. **b**—bandwidth. b. A schematic demonstration of top-frequency selectivity. Three different stimuli (3 shades of grey) having the same $f_t$ and varying $f_c$, all have a complete overlap with the excitatory band of the tuning curve and will consequently evoke the same response. Only a sufficiently narrow stimulus with the same $f_t$ (red) will evoke a weaker response—demonstrating that top-frequency selectivity is true for stimuli with some minimal bandwidth. c. Left: FBRAs of 3 example neurons. Middle and right: Tuning curves of the 3 example neurons shown on the left sorted by $f_t$ or $f_b$ of all stimuli. Black dots are data points corresponding to the firing rate at each stimulus. Red line is a fitted gaussian. The quality of fit for each tuning curve to a gaussian are shown on the top right of each plot. d. The quality of fit for each tuning curve when the tuning curve is plotted as top frequency (Y axis) versus bottom frequency tuning curves (X axis). Red line is the identity line. Q1 neurons are plotted separately (filled circles) to Q2-4 neurons (open circles). Inset: A bar graph showing $\Delta R^2 = (R_t^2 - R_b^2)$. Neurons from Q1 have significantly higher $R_t^2$'s (student's t-test, $p < 10^{-6}$).

observation, we calculated the quality of fit between these neuronal tuning curves and a Gaussian. As a negative control, we also calculated fits to the responses as a function of the bottom frequency—$f_b$, of the BBSs (Fig 5c, right). Neurons with asymmetric FBRAs (i.e. 'Q1' neurons in Fig 3d) showed better fits when their responses were plotted as a function of $f_t$ as compared to when plotted as a function of $f_b$ (Fig 5c—compare center to right). Neurons with symmetrical FBRAs showed similar fits to Gaussians in either condition (Fig 5c—bottom). Directly comparing the quality of these two fits, we found a tendency for the fits to be better as a function of $f_t$ than as a function of $f_b$ (Fig 5d), with most of the bias attributed to Q1 neurons (Fig 5d—inset). These data support the claim that this subgroup of neurons in ACx encodes the top frequency of broadband sounds.

## Discussion

### Protocol to study bandwidth tuning in ACx

Broadband sounds of varying bandwidths have been used to study single neuron responses throughout the auditory system [45–47], although their use in ACx is not very common

[38–40]. Using such sounds, it has been previously shown that neurons in ACx exhibit bandwidth tuning, often expressed as a non-monotonic response profile to bandwidth (see e.g., figure 1b in Li *et al.*, 2019 [40]). This type of response profile emerged from our data as well. For example, consider the FBRA in Fig 1e, and envision we would have played only the BBS around 8kHz (Fig 1e, column containing boxes 1–4). Importantly, we demonstrate here that limiting the exploration to stimuli centered on one frequency, as has been done previously in mice [39, 40] can render more complex response patterns undetectable. For example, by using the full 2D frequency landscape of center frequency and bandwidth, we found that the bandwidth which evokes the strongest response in a neuron (the optimal bandwidth for a given center frequency) could well be different around different central frequencies. Notably, the dependence of optimal bandwidth on center frequency followed clear regularities (Figs 2 and 3). These regularities were detected in a large group of responsive neurons in our dataset.

The protocol we used has a few merits. First, as our data show, we were able to readily measure two distinct phenomena—bandwidth tuning and sideband inhibition, which are tightly related. Indeed, bandwidth tuning in our hands is invariably a consequence of an inhibitory band in higher frequencies, accounting for the sensitivity for top edges. Second, since spontaneous firing rates in cortex are generally low, inhibition had a small effect over baseline firing rates and was therefore harder to detect when presenting only a small number of frequencies across the stimulus landscape. Of note, inhibition has already been shown to be made easier to detect by using two-tone presentations [48, 49]. However, since the number of two tone combinations to probe the full frequency landscape is very large, previous work often used only limited sets of pairs, based on individual characterization of each of the neurons recorded (e.g. one of the two tones is often selected to be at the preferred frequency of the neuron). Such choices limit again the conclusions that can be derived from such experiments (but see [50]). An additional fruitful approach that showed the existence of inhibitory sidebands in the tuning curves of ACx neurons is estimating their spectral-temporal receptive fields (STRF) [51–54]. The STRF is estimated by playing sound that are complex in both their spectral and temporal modulation and applying reverse correlation techniques to the neuronal response. Both our stimulus set and our model are simpler then the protocol needed for STRF estimation. However, STRFs are advantageous in pointing important spectral and temporal properties of neuronal tuning curves. Our data supplement and strengthen the STRF approach.

## Limitations of our study

Our findings have a few limitations. First, most neurons in our dataset are located in several sub-regions of the ACx, which deviates from previous work in the mouse, that focused on A1. In turn, this makes direct comparisons with other A1 studies more challenging [39, 40]. Second, neurons in our data were biased to respond to low frequencies. This bias may stem from the specific coordinates of the penetrations of our recording electrodes. The limited window of tonotopy from which we measured, prevents us from assessing how edge detection aligns with the tonotopic map. Third, our recordings were restricted to deep layers (L5/6) as opposed to previous studies that tested bandwidth-tuning which were restricted to more superficial layers or L4. For example, Li and colleagues [40] showed that bandwidth tuning is more prominent in L2/3 than in L4. The existence of bandwidth tuning already in L4 suggests that it might be found upstream to ACx. The extent to which the SS of L5/6 neurons reflects upstream mechanisms, or is determined by local interactions in ACx, remains unclear. Fourth, although BBS are slightly more complex than pure tone, they are still restricted stimuli. BBS are temporally static, and lack the complexity of natural stimuli. While we show that our model is generalizable to an additional set of static stimuli (i.e.- notches), we did not test our model's predictive

power over the responsiveness of ACx neurons to temporally richer, more natural stimuli. This remains an open question for future exploration. Fifth, even in cases where neuronal and model FBRAs had strong correlations, responses to pure tones (PT, sounds with 0 bandwidth) were usually poorly replicated by the model (see Fig 3a). The main reason for that is that in our protocol, sound intensity was increasing with bandwidth, leaving PT at low intensity, and thus frequently evoking a weak response even from highly active neurons. However, it is interesting that some neurons were highly responsive to PT with respect to broadband stimuli (e.g. Fig 2a) despite them having low intensity, suggesting an exceptionally strong effect of SS. Lastly, although our model explained well the activity of a large group of auditory neurons, it was nevertheless a poorer fit for others. This might be due to the fact that our model is simple, with only few parameters that attempt to describe only local rules of cortical connectivity. The exact shape of excitatory and inhibitory bands of biological neurons is likely more complex than our model can account for [55]. For example, top-down connections could modulate the local inhibitory landscape in ways that we did not explore here [56–58]. ACx neurons that do not fit the model could be involved in other auditory computations that are not captured by our model and, importantly, could be spectrally and/or temporally context-dependent [59–65].

## Mechanisms of SS in ACx

To explain FBRA profiles, we modified the Ricker wavelet function (also known as the Mexican hat) that is widely used to model SS in the retina [66]. Our modified version, the MRW, is a simple model with only 4 parameters that allows characterizing not only SS, but both the extent and symmetry of SS within a single dimension. Given the success of using MRW to study the extent and asymmetry of SS in ACx, it could be a useful tool to study SS in other systems as well. Based on the neuronal responses in ACx, the model shows that bandwidth tuning is largely a consequence of a tuning curve that has an excitatory frequency-band and single powerful inhibitory band above it. One mechanistic explanation that was suggested for this kind of asymmetry is local inhibition by somatostatin expressing interneurons. Specifically, somatostatin interneurons have been implicated in mediating neuronal suppression by side-band frequencies above the excitatory band [67], as well as explicitly in bandwidth tuning [39]. Interestingly, somatostatin interneurons have been shown to mediate SS in visual cortex as well [4], indicating further SS as a general cortical mechanism rather than modality specific [33]. Deciphering how asymmetrical SS is mediated by the precise structure of inhibition and excitation in the auditory system is a theoretical and experimental open challenge.

## Sound edges and beyond

We suggest that neurons in ACx with asymmetric SS behave as edge detectors, as they are particularly tuned to the top frequency of a BBS (Fig 5). Notably, edge detection by asymmetry is not a new idea, dating back almost 30 years ago. In fact, similar ideas have been suggested before from recordings in the ACx of anesthetized ferrets [48, 68] and awake owl monkeys [69] using different approaches. For example, Shamma and colleagues extracted the excitatory and inhibitory bands in the tuning curve of neurons using a two-tone protocol, showing that neurons with a single inhibitory band above the excitatory band prefer top spectral edges on their preferred frequency over bottom edges (and the opposite for neurons with the inhibitory band below the excitatory band) [48]. Fishbach and colleagues developed a model that accounted for those data, and showed that the model parameters showed smoother topographical mapping [68]. Our protocol supports and extends those earlier findings. First, we extend this idea to data in mice and to the ACx in the awake state. Second, using a large variety of

BBSs we show that neurons that respond to top edges are often much less sensitive to the bottom edge and the rest of the frequency content of stimuli, indicating a specificity of this type of computation. Third, the large variety of BBSs has allowed us to sort stimuli according to their top frequency and reveal a strong, unimodal tuning curve to that parameter, further indicating that spectral edge is likely a feature extracted by these neurons. Lastly, since we collected data from several cortical regions, our results suggest that SS exists in neurons beyond primary auditory cortex.

We speculate that edge detection may be useful for auditory scene analysis—the segregation and grouping of different spectral components of sounds [70]. The auditory system has been suggested to extract features such as harmonics [71], temporal patterns [72] and temporal and spectral edges [68, 73] for the inference of distinct components of the auditory scene. Indeed, responses to spectral edges such as those shown here may be an additional component of the computation needed for construction of auditory objects by the ACx.

An additional related finding was introduced by Zhang et al [34], which suggested that sideband inhibition takes part in neuronal selectivity to the directionality of frequency sweeps. Similar to the findings presented here, this paper showed that neurons in rat auditory cortex with a preferred frequency at the lower frequency range tend to have a sideband at higher frequencies (i.e. positive skewness). Additionally, they have shown that neurons with a preferred frequency at a higher frequency range have an inhibitory sideband at lower frequencies. Here we did not find many neurons with a lower inhibitory sideband (i.e. negative skewness), which might imply a species/ strain specific property or be a result of our data being biased to neurons preferring low frequencies. Finally, using conductance measurements, Zhang and colleagues [34] have shown that inhibition is delayed with respect to excitation which together with the spectral asymmetry, results in a frequency sweep directionality preference.

Finally, we note that the response profile which we measured might only be a particular case of a more general computation. The response profile of top-edge detecting neurons (i.e. Q1 neurons) is qualitatively similar to the first derivative of a gaussian (i.e. g1 wavelet). In this context, Fishbach and colleagues[73] have built a spectro-temporal model for auditory responses, whereas neurons such as those we call 'Q1 neurons' compute the first spectral derivative of sounds. Indeed, it has been shown that convolving a signal with a g1 wavelet gives back the signal's (smoothed) first derivative [42]. According to this interpretation, BBS with sharp edges such as those we presented here, will evoke the strongest response, with sharp edges having infinite derivatives. Thus, we speculate that neuronal responses in ACx will be proportional to the (absolute) magnitude of the spectral derivative of a specific frequency.

To conclude, using a single, generic protocol and a simple mathematical model we tap onto three phenomena in single ACx neurons: bandwidth tuning, sideband inhibition and edge detection (Figs 1–3 and 5). Further, we show that a significant group of neurons in awake mouse ACx show strong tuning to sound edges (Fig 5). The simplicity of this stimulus set could be exploited in future research to elucidate the contribution of different auditory brain regions to edge detection, and perhaps be used in behavioral experiments to see how such computation translates to perception.

## Materials and methods

### Ethics statement

We used 8 to 12-week-old C57/b6 female mice (n = 5). All experiments were performed in head-fixed awake mice and approved by the Institutional Animal Care and Use Committee of the Hebrew University of Jerusalem (NS-18-15521-4).

### Extracellular recordings using Neuropixels

We used Neuropixels [74] for extracellular recordings of spiking activity in left ACx. Prior to the recording, a custom-made metal bar was implanted on the mouse skull, and a small craniotomy was made on the left hemisphere (coordinates relative to bregma: anterior 2.5 mm, lateral 4.2 mm). The craniotomy was protected by a wall of dental cement and covered directly with a silicone elastomer (WPI; Kwik-Cast catalog #KWIK-CAST). The surgery was done under Isoflurane anesthesia (2%) and a subcutaneous injection of carprofen (0.004 mg/g). Mice were given 1–2 d to recover. On the day of the recording, animals were head-fixed and the craniotomy was exposed. Then, a Neuropixels probe 1.0 (IMEC, phase 3A) was inserted through the craniotomy in a 20–30 degree tilt (from vertical position) and lowered into the brain, 3 mm deep. Penetration and probe depth were performed and monitored using a micromanipulator. Probes were covered with a fluorescent dye [DiI (Invitrogen catalog #V22885) or DiO (Invitrogen catalog #V22886)] before penetration, to enable reconstruction of the penetration sites in high resolution. In each mouse, we performed one or two consecutive probe penetrations. Probe trajectories were reconstructed from consecutive coronal slices as detailed in earlier work [43, 62] and using an open source software ([75]; Fig 1D). Following the reconstruction, probe channels were annotated to the corresponding brain regions they were recording from (see Table 1).

All recordings were acquired using Neuropixels 1.0 (IMEC, phase 3A), a base-station connector (IMEC) and a National Instruments chassis with an IMEC slot. An external reference electrode (Ag/AgCl wire) was positioned on the skull and submerged in saline. Data were sampled at 30 kHz, with action potential band filtered to contain 0.3- to 10kHz frequencies. Action potential band gain was set to 500. All recordings were automatically sorted using the algorithm of Kilosort 2, with its default parameters (e.g. central part of the autocorrelation—2ms; contamination—20%) ([76] https://github.com/MouseLand/Kilosort2). Following automatic sorting, manual sorting was performed using the "Phy 2.0" open-source GUI (UCL; https://github.com/cortex-lab/phy). During manual sorting, spike clusters were merged based on assessment of wave-form similarity and the appearance of drift patterns. Each spike cluster was assessed by criteria of waveform size, consistency of firing rates, the presence of short-latency inter-spike intervals, auto and cross correlograms, and principal component analysis. If a cluster was classified by Kilosort as a well isolated SU with minimal contamination (based on violations of the expected refractory period, see https://github.com/MouseLand/Kilosort/wiki), and the manual inspection showed it to be satisfactory on all above mentioned accounts, it was tagged as a SU. If a cluster did not meet the above criteria, it was excluded from analysis. A more detailed explanation for the manual sorting guidelines can be found in https://phy.readthedocs.io/en/latest/sorting_user_guide. Fig 1D was generated using the Allen CCF software (UCL; https://github.com/cortex-lab/allenCCF).

### Auditory stimuli

Sound stimuli were presented through a free field speaker (ES1; TDT) positioned 5cm from the animals' right ear and were delivered to the speaker at 500 kHz sampling rate via a driver (ED1; TDT). Stimuli consisted of pure tones (either 11 or 17 different frequencies starting from 4 kHz and going up to either 22.6 or 64kHz in 25% octave intervals). BBSs had center frequencies matching the pure tone frequencies with either 10 or 11 different bandwidths linearly spaced between 5% and 100% octaves (full width) around the center frequency. Notice that since the steps in the frequency axis and in the bandwidth axis are different (25% vs. ~10%), FBRAs have a step-like shape that is not a result of neuronal/model properties, but merely the choice of stimuli. Notch stimuli were inverse to BBSs with a minimal frequency of 2kHz and a

maximal frequency of a full octave above the highest pure tone frequency played at that protocol. Both BBSs and notches were created by summing pure tones of all frequencies within the bandwidth spaced 0.1% octave apart, with an equal amplitude and randomized phases. Pure tone sound levels were ~55 dB SPL, while 1 octave BBS sound level were ~85 dB SPL. Sound levels for notches were similar regardless of notch bandwidth and measured ~90 dB SPL. All stimuli were 100 ms in duration, with 5 ms on and off linear ramps. Each pure tone was repeated 10 times, and each broadband stimulus was generated 10 times since there is a random component to its creation. All stimulus repetitions were presented in a random order with 1 second inter-stimulus intervals.

## Data analysis

Data analysis and statistics were performed using custom written code in MATLAB (MathWorks). Analysis was restricted to neurons from the following auditory regions—AUDp, AUDv, AUDd and TeA, and only to those SU's that were significantly excited by BBSs in their ON-response. A significant excitatory response was determined by performing a one-tail paired students' t-test between the number of spikes during stimulus presentation (100 ms) and the number of spikes during the 100 ms immediately before the stimulus (pre-stim window), Bonferroni corrected for multiple comparisons. Neurons that were inhibited by BBS, strictly OFF-responsive to BBS or responsive only to notch stimuli were not analyzed further (see S1 Fig). PSTHs were calculated for 10 ms bins. Normalized amplitudes for FBRAs were taken as the mean difference in firing rate during stimulus presentation and the pre-stim window.

## Modulated Ricker wavelets

MRWs obeyed the function:

$$MRW(f) \, \alpha \, (w^2 + {}^I/_E \cdot (f - \mu)^2) \cdot e^{-\frac{(f-\mu)^2}{2w^2}} \cdot \phi^*(f, \mu, w, s)$$

$$\phi^*(f, \mu, w, s) = (0.5 + 0.5 \cdot s) \cdot \phi\left(f, \mu, \frac{w}{2}\right) + (0.5 - 0.5 \cdot s) \cdot \left(1 - \phi\left(f, \mu, \frac{w}{2}\right)\right)$$

Such that $\phi\left(f, \mu, \frac{w}{2}\right)$ is a Gaussian cumulative distribution function with mean $\mu$ and standard deviation $\frac{w}{2}$, which was chosen heuristically for not producing an artifact. Wavelets were normalized such that their maximal value is 1 (Fig 2b, S2a Fig, S1 Video). MRWs were represented as vectors of weight values across the frequency axis. For calculation of model-FBRAs, stimuli were represented as vectors of intensity values in frequency space. To account for the phenomenon of cochlear widening [77], broadband stimuli, which initially were a rectangular pulse of height 1, were concatenated with a linear increase/decrease from half an octave away on either side (S2a Fig). This simple widening was chosen in order not to achieve too many degrees of freedom over the model. This was performed similarly for notch stimuli (S2b Fig). Finally, to calculate a model neuron response to each stimulus, the inner product of the stimulus vector and the MRW went through a positive rectifier to account for the low spontaneous firing of ACx (S2a Fig).

## Model fitting

To fit model FBRAs to neuronal FBRAs, we used a modification of the Matlab function *fminsearch()*, that allows restricting parameter values. The value to minimize was the negative of the Pearson correlation between neuronal and model FBRAs, (excluding responses to pure tones) + 0.05 · **w**, in order to avoid unrealistically large values of **w**. **μ** was restricted to the

range of presented frequencies. $w$ was restricted to widths above 5% of an octave. $^I/_E$ was restricted between 0 and 1. $s$ was restricted between -1 and 1. For fitting shuffled data, the FBRA of each neuron was shuffled before fitting. Shuffling was done by randomizing the location of each of the 121 mean responses, along the vector of central frequencies and bandwidths. Data shuffling was performed 100 times on neuronal FBRAs before fitting the model and then averaged. Shuffle correlation values in Figs 3b and 4d are the averages. For fitting neuronal responses according to $f_t$ or $f_b$ to a gaussian, we used the Matlab function fit().

## Supporting information

**S1 Fig. Neuronal counts.** a. A scheme showing the number of SUs outputted by the spike sorting procedure (n = 577), the number of neurons responsive to one of various attributes of sounds (n = 451), highly responsive units to BBSs (n = 149), and the number of neurons with high fit-correlations (above median) to the MRW model (n = 74). see Methods for further information.
(PDF)

**S2 Fig. Response function.** a. Schematic of the calculation of a response of a model neurons with a given tuning curve (middle) to an example BBS modeled with cochlear widening (right). The inner product of the tuning curve and the stimulus is then positively rectified. b. Example notch stimulus, modeled with cochlear widening.
(PDF)

**S3 Fig. Additional examples of model fitting.** a. Top: FBRAs of 4 neurons from 3 mice. Middle: model FBRAs fitted to the same neurons. Bottom: fitted MRWs of the same neurons.
(PDF)

**S1 Video. The Modulated Ricker Wavelet model and resultant FBRA.** Left: An MRW tuning curve of a model neuron with all parameters values stated in red in the top right. The shape of the tuning curve is changing by changing the parameter values. Right: A synthetic FBRA of the model neuron.
(MP4)

**S2 Video.** Left: An MRW tuning curve of a model neuron with $^I/_E = 1$, $s = 1$ (i.e.—Q1 neuron). The transparent grey rectangle represents a BBS. The rectangle on the top right fills up with red color the more the model neuron responds to the BBS, such that a fully red rectangle corresponds to a maximal response. BBS with the same top frequency evoke similar responses as long as the stimulus is wider than the excitatory band of the tuning curve. BBS with the same bottom frequency do not necessarily evoke the same response.
(MP4)

## Acknowledgments

We thank Yoav Rubinstein and members of the Mizrahi laboratory for comments on the manuscript. We thank Itai Pomson for technical help.

## Author Contributions

**Conceptualization:** Omri David Gilday, Adi Mizrahi.

**Data curation:** Omri David Gilday, Benedikt Praegel, Ido Maor.

**Formal analysis:** Omri David Gilday.

**Funding acquisition:** Adi Mizrahi.

**Methodology:** Omri David Gilday, Tav Cohen, Israel Nelken.

**Supervision:** Adi Mizrahi.

**Writing – original draft:** Omri David Gilday, Adi Mizrahi.

**Writing – review & editing:** Israel Nelken, Adi Mizrahi.

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
