## [Decision Letter · Decision Letter 0]

8 Nov 2022

Dear Prof Mizrahi,

Thank you very much for submitting your manuscript "Surround Suppression in Mouse Auditory Cortex Underlies Auditory Edge Detection" for consideration at PLOS Computational Biology. As with all papers reviewed by the journal, your manuscript was reviewed by members of the editorial board and by several independent reviewers. The reviewers appreciated the attention to an important topic. Based on the reviews, we are likely to accept this manuscript for publication, providing that you modify the manuscript according to the review recommendations.

Dear Adi and co-authors,

I hope that you will find the feedback of two expert reviewers useful for your revised manuscript. I have also found your paper quite interesting and well done. The two populations of neurons that you found using this approach is intriguing as well. At this stage of the review, I have just two major comment and a few minor ones. My first major comment is that in the limitations of your study you never mention that you have not attempted to generalize it to stimuli beyond short static (and “non-natural” ) stimuli. If your description of the “tuning” of the neuron is correct it should be able to predict time-varying responses to natural sounds as the mouse experiences it in its environment. I agree with you that a family of broad band stimuli is better than short pure tones or simple combination of tones but it is still quite a restricted stimulus. You should at least discuss these limitations. My second major comment is that you are effective obtaining static spectral receptive fields using a parametrized stimulus. You briefly mention the limitation of ignoring the temporal dimension (citations 57, 69) but you might also state that inhibitory side bands have been reported before by estimating spectral temporal receptive fields (STRFS) both in mammals and in birds. In these studies complex broad band sounds with spatial temporal structure are used in combination with reverse correlation techniques. One could argue that that “older” approach is even better? Some missing relevant citations: Qiu A, Schreiner and Escabi J. Neurophys 2003; Miller et al. J. Neurophys 2002; Depireux et al. 2001; Woolley et al. J. Neuroscience 2009.

Minor:

1. Top of p.6: When describing the stimuli and the effect of increasing the bandwidth, I would say that the signal rms increased. If you talk about sound intensity, you would have to define what you mean by intensity and clearly distinguish it found sound loudness.

2. Bottom of p.7. You say: “These data show that our model has a strong descriptive power over the electrophysiological data.” You are actually make correlations between one model - (model FBRA) - and one summary statistic of your ephys data - emperical FBRA. You are not predicting time-varying responses or the occurrence of single spikes. Even for 100 ms long static stimuli, there is clearly some dynamics in the response.

3. The notch stimuli are a good control. To be candid and really generalize, one might have also used short natural stimuli (100 ms) and in a different classes of sounds all together such as mouse song.

4. It might fine to use the default values in kilosort for selecting SU but I would state more explicitly what algorithm was used and with what parameters. The central vs shoulder part of the auto-correlation function that is estimated in those algorithms to assess refractory violations might not be appropriate for your data.

Sincerely,

Frédéric E. Theunissen

Academic Editor

PLOS Computational Biology

Daniele Marinazzo

Section Editor

PLOS Computational Biology

Dear Adi and co-authors,

I hope that you will find the feedback of two expert reviewers useful for your revised manuscript. I have also found your paper quite interesting and well done. The two populations of neurons that you found using this approach is intriguing as well. At this stage of the review, I have just two major comment and a few minor ones. My first major comment is that in the limitations of your study you never mention that you have not attempted to generalize it to stimuli beyond short static (and “non-natural” ) stimuli. If your description of the “tuning” of the neuron is correct it should be able to predict time-varying responses to natural sounds as the mouse experiences it in its environment. I agree with you that a family of broad band stimuli is better than short pure tones or simple combination of tones but it is still quite a restricted stimulus. You should at least discuss these limitations. My second major comment is that you are effective obtaining static spectral receptive fields using a parametrized stimulus. You briefly mention the limitation of ignoring the temporal dimension (citations 57, 69) but you might also state that inhibitory side bands have been reported before by estimating spectral temporal receptive fields (STRFS) both in mammals and in birds. In these studies complex broad band sounds with spatial temporal structure are used in combination with reverse correlation techniques. One could argue that that “older” approach is even better? Some missing relevant citations: Qiu A, Schreiner and Escabi J. Neurophys 2003; Miller et al. J. Neurophys 2002; Depireux et al. 2001; Woolley et al. J. Neuroscience 2009.

Minor:

1. Top of p.6: When describing the stimuli and the effect of increasing the bandwidth, I would say that the signal rms increased. If you talk about sound intensity, you would have to define what you mean by intensity and clearly distinguish it found sound loudness.

2. Bottom of p.7. You say: “These data show that our model has a strong descriptive power over the electrophysiological data.” You are actually make correlations between one model - (model FBRA) - and one summary statistic of your ephys data - emperical FBRA. You are not predicting time-varying responses or the occurrence of single spikes. Even for 100 ms long static stimuli, there is clearly some dynamics in the response.

3. The notch stimuli are a good control. To be candid and really generalize, one might have also used short natural stimuli (100 ms) and in a different classes of sounds all together such as mouse song.

4. It might fine to use the default values in kilosort for selecting SU but I would state more explicitly what algorithm was used and with what parameters. The central vs shoulder part of the auto-correlation function that is estimated in those algorithms to assess refractory violations might not be appropriate for your data.

Reviewer's Responses to Questions

**Comments to the Authors:**

Reviewer #1: The manuscript entitled ‘Surround Suppression in Mouse Auditory Cortex Underlies Auditory Edge Detection’ is truly very interesting. The authors record from the auditory cortex of awake mice using Neuropixels probes and a novel and very elegant sound stimulation protocol. The combination with modelling, allows them to explore the sensitivity of auditory neurons to the frequency bandwidth of the stimulus. The authors find that neurons are sensitive to the bandwidth of the sound stimulation, that this sensitivity is evidenced through surround suppression, and that the asymmetry of this side inhibition results in the coding of upper frequency edges. Surround suppression and edge detection are phenomena well explored in other modalities, notably in vision, and implied in auditory data. The current protocol presents evidence for edge detection in the auditory cortex and opens a door to study this dimension of auditory processing. The experiment is well designed, and the interpretation of the results is justified, although I miss details in the analysis of edge detection. The manuscript is well written, very clear, and the figures generally appropriate, although see comments below. I found the videos a wonderful contribution to the clarity.

Main comments:

1) Edge detection is introduced only in Figure 5 and then hardly explored. The shift leftwards in tuning with increased bandwidth is evident already in Figure 1 and leaves the reader wondering why the relationship between bandwidth (and edges) is not explored in more detailed there. The edge detection could be brought forward, already to figure 2 or 3. I think the authors should decide. If they want to leave edge detection to Figure 5, it would help the reader to have, in Figure 1, some indication of how the bandwidth relates to the frequency shift, for example by illustrating for one fc the frequency ranges that are covered with each bandwidth. Then, in figure 5, a transition between the BFRAs and the tuning according to ft is necessary. For example one could rearrange the FBRAs such that rows are ft.

2) I miss a more thorough analysis of the relationship between the 2 variables identified as more informative, ‘s’ and ‘I/E’, and the other variables. This could include a quantification of the distribution of responses-patterns. I miss more examples of neurons with low model correlation, or at least at the edge of the threshold.

3) I wonder how the distribution of edge frequencies in the population contributes to the tonotopic map. Would a tonotopic map centered built with responses to pure tones, look very different form one built with top edge frequencies. If so, how? If the recordings do not span a wide enough range of the tonotopic map, the authors could discuss the issue.

Minor comments:

-Line numbers are missing

Abstract:

-What is the question?

Methods:

-Were spikes quantified only as ‘the number of spikes during stimulus presentation (100

ms)’?

-In the SU validation, what is the minimum mean spike amplitude, number of spikes per protocol, ISI?

Results:

-In most examples, throughout the manuscript, sounds with zero bandwidth elicit very weak responses. I am not sure if these sounds are what is described as pure tones in the methods and, therefore, are sounds of lower intensity? If so, are they necessary in the plots? An exception is the one shown in Figure 2a top right, which also happens to not show responses at increasing bandwidth. Please comment.

-Figure 1, the asymmetrical FBRAs have a stepwise pattern. Is this because of a difference between the resolution of the fc (plotted) and the frequency range increase with bandwidth?

-Figure 3, examples of low correlation neurons?

-Figure 3c is not very informative since neuron clusters fall on a very small region of 1 or 2 quadrants. A histogram of the projection of the each of the 2 axes in Figure 3b would be better in my view. In fact, reading further, I would move Sup. Fig 3 to the main figures. See also main comment 2 above. I like that the dots with -1 I/E values fall on the left of the ordinate, as if they were embarrassed.

-Was there any correlation between ‘s’ or ‘w’ values and best frequency. Do either of them increase or decrease with BF? The distribution is very skewed so probably not.

-What is the value of the population median in ‘we focused on SUs that had the highest correlations with the model, defined as being above the population median’? I could not find it. Since the distribution of correlations is flat, what does this threshold mean?

-Figure 4e, the fit might be strong but most of the nFBRAs are very close to zero and only as the FBRA becomes 0.6 or above does the nFBRA start to increase. Which brings me back to the question of what is a meaningful threshold for the FBRA correlation. I did not find this analysis very useful. Also, when one then reads about the edge detection in the next section, one is left wondering how the notch relates to this.

-Figure 5c. It would help the explanation, and to keep in line with the message of 5a and 5b, if, in addition to the FBRA and tuning curves, there would be also an FBRA aligned by the top, rather than current, frequency. Maybe a video, showing how the tuning changes as the FBRA shifts alignment from fc to ft?

-Figure 5d is interesting but I would like to see a correlation between Rt and other variables found of meaning before, such as ‘s’ and ‘I/E’, or the model correlation. If pure tone responses were comparable, it would be interesting to see the relationship between Rt and R pure tone.

-While the responses of a neuron to a given set of bandwidths might be the same when these are aligned by upper edge frequency, the maximum response will be to a more centered frequency (t would be nice to see examples of this on the manuscript). Is this because these centered frequencies are played at higher intensities? Or does this reflect a combined sensitivity to edge and center.

Discussion:

-First paragraph: rephrase? It seems to go back and forth. The group of sentences starting ‘Importantly,’ seems to repeat the previous ones. Also, the sentence starting ‘Notably’ is, I think misleading. I did actually miss the analysis of relationship of effects with center frequency…

-Second paragraph: I appreciate the humbleness, but did you really mean ‘has few merits’? or rather ‘has a few merits’? Same, but less humble, with the limitations.

-Please discuss the stepwise pattern in asymmetric FBRAs

-Please discuss the relationship with tonotopy.

Reviewer #2: In this study, Gilday et al use the Ricker-wavelet model (Mexican-hat) to parsimoniously explore bandwidth tuning, sideband inhibition and edge detection in the auditory cortex of awake mice. The authors use a 2D stimulus space to investigate the interactions of center frequency and bandwidth. Based on extracellular recordings across several auditory fields, the authors show a variety of complex trends in frequency-bandwidth response areas (FBRAs) from a population of neurons. To elaborate on some salient features of the data, they use the Ricker-wavelet framework to model FBRAs by manipulating 4 parameters. One of the limitations of the Ricker-wavelet model implemented in the study is that it assumes neural architectures that do not fit with current knowledge of cortical connectivity. As a result, the field of computational vision has largely moved on from the Ricker-wavelet model in order to incorporate recent discoveries of visual cortical architecture and function (e.g. Angelucci et al, Annu Rev Neurosci 2017). The Gilday et al study could potentially be a resource to auditory computational neuroscientists if it incorporated a more modern biological foundation, including new findings on auditory connectomics and function. Another suggestion to enhance the impact of the study would be to relate the findings to functional maps of the areas they are studying (e.g. auditory subfields, tonotopy, etc). Lastly, I suggest that the authors make a greater effort to discuss how their work extends what is already known about surround suppression in the auditory cortex, in particular Zhang et al 2003 (Science). Using conductance measurements of synaptic input in vivo, that study provided numerous insights on side-band inhibition, asymmetries and how they relate to functional maps.

**Have the authors made all data and (if applicable) computational code underlying the findings in their manuscript fully available?**

Reviewer #1: **No: **

Reviewer #2: None

PLOS authors have the option to publish the peer review history of their article (what does this mean?). If published, this will include your full peer review and any attached files.

Reviewer #1: No

Reviewer #2: No

Figure Files:

Data Requirements:

Reproducibility:

References:

---

## [Editor Report · Decision Letter 1]

9 Jan 2023

Dear Prof Mizrahi,

We are pleased to inform you that your manuscript 'Surround Suppression in Mouse Auditory Cortex Underlies Auditory Edge Detection' has been provisionally accepted for publication in PLOS Computational Biology.

Best regards,

Frédéric E. Theunissen

Academic Editor

PLOS Computational Biology

Daniele Marinazzo

Section Editor

PLOS Computational Biology

Dear Adi,

Thank you for addressing all the (mostly) minor comments for the reviewers. Congrats on a nice study.

Frederic.

---

## [Editor Report · Acceptance letter]

14 Jan 2023

PCOMPBIOL-D-22-01350R1 

Surround Suppression in Mouse Auditory Cortex Underlies Auditory Edge Detection

Dear Dr Mizrahi,

I am pleased to inform you that your manuscript has been formally accepted for publication in PLOS Computational Biology. Your manuscript is now with our production department and you will be notified of the publication date in due course.

With kind regards,

Zsofia Freund
